Oyster reef restoration supports increased nekton biomass and potential commercial fishery value

Humphries Austin T. 1 2 4 austin.humphries@gmail.com
La Peyre Megan K. 3 mlapeyre@agcenter.lsu.edu
1 School of Renewable Natural Resources, Louisiana State University, AgCenter , Baton Rouge, LA , USA
2 Atlantic Ecology Division, United States Environmental Protection Agency , Narragansett, RI , USA
3 United States Geological Survey, Louisiana Fish and Wildlife Cooperative Research Unit, School of Renewable Natural Resources, Louisiana State University AgCenter , Baton Rouge, LA , USA
4 Current affiliation: College of the Environment and Life Sciences, University of Rhode Island , Kingston, RI , USA
Esteban María Ángeles
Electronic publication date: 2015 Aug 25
Publication date: 2015
Volume: 3
Electronic Location ID: e1111
Received 2015 Apr 21; Accepted 2015 Jun 27
Copyright year: 2015
License: This is an open access article, free of all copyright, made available under the Creative Commons Public Domain Dedication. This work may be freely reproduced, distributed, transmitted, modified, built upon, or otherwise used by anyone for any lawful purpose.
License URL: https://creativecommons.org/publicdomain/zero/1.0/

Keywords: Ecological valuation, Biogenic habitat, Estuarine ecology, Gulf of Mexico, Essential fish habitat, Ecosystem engineer, Fisheries, Living shorelines, Facilitation

Funding: Louisiana Department of Wildlife and Fisheries The Louisiana Department of Wildlife and Fisheries provided financial support. The funders had no role in study design, data collection and analysis, decision to publish, or preparation of the manuscript.

==============================
Across the globe, discussions centered on the value of nature drive many conservation and restoration decisions. As a result, justification for management activities increasingly asks for two lines of evidence: (1) biological proof of augmented ecosystem function or service, and (2) monetary valuation of these services. For oyster reefs, which have seen significant global declines and increasing restoration work, the need to provide both biological and monetary evidence of reef services on a local-level has become more critical in a time of declining resources. Here, we quantified species biomass and potential commercial value of nekton collected from restored oyster (Crassostrea virginica) reefs in coastal Louisiana over a 3-year period, providing multiple snapshots of biomass support over time. Overall, and with little change over time, fish and invertebrate biomass is 212% greater at restored oyster reefs than mud-bottom, or 0.12 kg m−2. The additional biomass of commercial species is equivalent to an increase of local fisheries value by 226%, or $0.09 m−2. Understanding the ecosystem value of restoration projects, and how they interact with regional management priorities, is critical to inform local decision-making and provide testable predictions. Quantitative estimates of potential commercial fisheries enhancement by oyster reef restoration such as this one can be used directly by local managers to determine the expected return on investment.

Introduction

Millions of people depend on oysters and the reefs they build for income and livelihood (Kirby, 2004). While oysters are valued as a fishery commodity, they also provide important ecosystem services, including water quality improvement, shoreline protection, and invertebrate and fish habitat (Breitburg et al., 2000; Cressman et al., 2003; Coen et al., 2007; La Peyre et al., 2014). Nevertheless, reports have indicated a significant global decline in oyster habitat over the last century, with greater than 85% of historical reefs functionally impaired (Beck et al., 2011). Regional assessments in the US have quantified concomitant loss of oyster biomass to declines in water quality over the same time frame (Zu Ermgassen et al., 2013), and calculated the potential value of water filtration and nutrient sequestration provided by oyster reefs (Kellogg et al., 2013; Pollack et al., 2013). Recent work on the role of fringing oyster reefs in shoreline protection suggests reefs may contribute to reduced shoreline retreat rates during high energy storm events (La Peyre et al., 2014). Attempts to relate invertebrate and fish biomass to overall reef extent or characteristics have resulted in contrasting results (Rodney & Paynter, 2006; Humphries et al., 2011). General consensus, however, is that understanding the potential role of oyster reefs in enhancing invertebrate and fish habitat requires improving our predictive capacity, and better understanding local context (Lenihan & Peterson, 1998; Lenihan, 1999; Grabowski et al., 2005).

The net effect of any oyster reef on fish and invertebrate biomass may be highly dependent upon the extent to which the ecological role played by the reef habitat is not redundant in the area (Walker, 1992; Grabowski et al., 2005; Geraldi et al., 2009). Specifically, to what extent do the oyster reefs provide unique recruitment habitat, or spatial refuge from predation, all of which may augment the prey base, increase foraging efficiency for predators, and enhance future productivity (Hixon, 1998; Syms & Jones, 2000)? Additionally, the long-term effectiveness of oyster reefs to provide ecosystem services is influenced by biological factors such as salinity and water temperature, both of which affect oyster recruitment and growth, or a reef’s ability to be self-sustaining and keep up with local sedimentation and subsidence rates (Southworth et al., 2010; Casas, La Peyre & La Peyre, 2015). This context-dependency is why local values are critical if we want to inform decision-making and understand how restoration efforts may generate benefits supporting local coastal communities.

Broad, large-scale studies are critically valuable in contributing to national and international discussions on the value of restored ecosystems; local studies, however, may lead to more support for local management activities by better framing the issue (Scannell & Gifford, 2013; Wiest, Raymond & Clawson, 2015). Furthermore, with broad-scale studies, the oftentimes large range in dollar estimates stresses the inherent variation or uncertainty in values, and is thus difficult to apply to local projects. For example, a recent review of ecosystem services provided by restored oyster reefs within the US compiled quantitative data on water quality services, fish habitat provision, and erosion protection values, and estimated reefs to provide services valued between $5,500 and $99,000 per hectare per year (Grabowski et al., 2012). Until we explicitly understand what drives these large ranges of values, the need for local assessments of ecosystem provisions is critical and will help to identify where restoration activities will add value.

In this study, we quantify biomass of fishes and invertebrates at six restored oyster reefs in a Louisiana estuary. We use species-specific biomass data to provide a snapshot of biomass support, and estimate enhanced value of commercial fisheries associated with reefs through comparisons with reference sites. As value is subjective, more regional level estimates are critical in assessing ecosystem services, which often represent the practical scales at which management strategies are designed and implemented.

Materials and Methods

Ethics statement

We obtained all necessary permits for the described study and operated under a scientific collection permit from the Louisiana Department of Wildlife and Fisheries to Dr. Megan La Peyre (S-03-2009, S-105-OYS-2010). No endangered or protected species were collected during this project. Furthermore, fish and invertebrates were collected under the Institutional Animal Care and Use Committee permit 08-005 to Dr. Megan La Peyre through the Louisiana State University Institutional Animal Care and Use Committee.

Study site

The study was conducted at Caillou (Sister) Lake, located in Terrebonne Parish, Louisiana (29°14′11.09N, 90°55′16.48W) (Fig. 1). Sister Lake is primarily an open water, brackish system with a mean tidal range of 0.3 ± 0.03 m (±one standard error) (LDWF/USGS recorder #07381349). Water levels are driven primarily by wind events; dominant winds are typically from the southeast, except during the winter when northerly winds accompany cold fronts.

Figure 1 Study area map.

Location of study area in Sister (Caillou) Lake, Terrebonne, Louisiana. Study regions are classified as North, West, or South. Two sites are displayed for each region, where paired oyster reef (25 × 1.5 × 1 m) and reference mud-bottom were sampled quarterly from 2009 through 2011.

We selected three regions (North, South, West) within Sister Lake for this study (Fig. 1). Within each region, we chose two sites for reef restoration (#1, #2), each with a paired reference mud-bottom treatment (200 m apart). Oyster reefs (25 × 1.5 × 1 m) were constructed in March 2009 with shucked, unaggregated oyster shell (La Peyre et al., 2014). All reefs were placed as close to the shoreline as possible (5–10 m) and were intertidal; however, due to the low tidal amplitude and water depth within our study area, the reefs were exposed less than 20% of the time over the study period (La Peyre et al., 2014).

Sampling procedure

Water quality variables were taken at each site concurrent with fish and invertebrate sampling which occurred quarterly for 2.5 years, beginning in June of 2009 and ending in October of 2011 (2009 = 3 sampling events; 2010 = 4 sampling events; 2011 = 3 sampling events). At each site, we measured temperature (°C), salinity, and dissolved oxygen (mg L−1). Water clarity (cm) was also measured using a Secchi disc at each sampling event.

To characterize fish and invertebrate assemblages, we used a combination of sampling gears including a gill net, bag seine, and plastic substrate trays. We used this range of gears in order to best characterize the entire community that may use the restored oyster reefs. The gill net was intended to characterize the transient fish and invertebrates, and can be defined as those individuals with large home ranges that may feed on or near oyster reefs (Breitburg, 1999). The bag seine was aimed at catching facultative residents that are associated with, and use, the areas adjacent to oyster reefs for either foraging or proximity to refuge. Substrate trays were buried and placed within the shell matrix and intended to quantify resident species, or those living within the shell matrix for refuge or nursery habitat.

At each treatment, a gill net (50 × 1.75 m with 5, 7, 10, 12, 14 cm monofilament sections) was first deployed to form a semicircular enclosure with the shoreline surrounding the treatment (oyster reef or mud-bottom). The gill net set times averaged 2 h, during which seine and tray sampling occurred. A bag seine (5 × 2 m with 3 mm square delta mesh) was then swept parallel to the shoreline, over mud-bottom, for a distance of 25 m. One sweep was executed on the ‘shore’ side of the treatment, between the treatment and salt marsh vegetation, while one sweep was executed on the “estuary” side of the treatment. Seines were taken to shore where collected nekton were removed and placed in labeled bags on ice for identification in the laboratory.

During reef construction, plastic substrate trays (63 × 52 × 11 cm) lined with 0.5 mm mesh screening were randomly placed in each oyster reef (6 oyster reefs ×3 substrate trays ×10 planned sampling events = 180 trays; sampling without replacement) and at mud-bottom treatments (6 mud-bottom treatments ×3 substrate trays = 18 trays; sampling with replacement). Substrate in the trays matched that of the reef (oyster shell) or reference (mud-bottom) treatments. Plastic substrate trays at oyster reef treatments were randomly sampled by quickly lifting the trays and placing the contents in 3-mm mesh bags. Substrate tray contents were rinsed to remove excess sediment by sieving tray contents on site, and contents placed in labeled bags on ice for identification in the laboratory. After substrate tray removal, substitute oyster cultch was used to fill the hole that was created by removing the tray. Substrate trays at reference treatments (mud-bottom) were anchored in the sediment using PVC poles and sampled with replacement.

After seine and tray samples were taken, the gill net was removed and all nekton were identified, weighed to the nearest 10 g (wet weight), and total length (TL) measured to the nearest 1 cm before being released on site. In the laboratory, nekton from seine and substrate tray samples was identified to species or the lowest practical taxon. Individuals of a species in each sample were weighed to the nearest 0.1 g (wet weight) and measured to the nearest centimeter of TL for fishes and shrimps, or carapace width for crabs. Thirty individuals were randomly subsampled to obtain lengths and weights of individuals from abundant species.

Nekton biomass and commercial fishery value calculations

Fish and invertebrate biomass were calculated using the species-level biomass totals from across gear types. Biomass totals at each sample site were divided by that particular reef’s area. The same area value that was used to represent the oyster reef was also used for the paired mud-bottom treatments. For seine biomass calculations, both seine pulls (in front of, and behind the reef or mud-bottom treatment) were summed to represent a total for that particular treatment area. Tray values were calculated by dividing the biomass of each tray by the area of each tray (0.3276 m2), then scaled to 1 m2. Community-level values were derived by summing all species-level biomass at a particular oyster reef or mud-bottom treatment within each shoreline, and then mean and standard error values were calculated from these values to get a sample event (seasonal) mean biomass (kg) per 1 m2. We calculated enhanced biomass as reef biomass minus mud-bottom biomass within each sample period.

Potential commercial fishery value was calculated by multiplying nekton biomass values by the commercial price for a particular species across gear types. Fish and invertebrate prices were derived from the National Marine Fisheries Service commercial landings online database and represented in 2011 US Dollars (http://www.st.nmfs.noaa.gov/commercial-fisheries/commercial-landings/annual-landings-with-group-subtotals/index). Species-level commercial fishery values were calculated for mud-bottom and oyster reefs at the sample level, and enhanced values were calculated the same as stated above for nekton biomass (as reef values minus mud-bottom values). Community-level numbers for commercial fish value were derived as described above for fish biomass, by summing all species-level values at treatments within each shoreline, then mean and standard error values calculated to get a sample event (seasonal) mean commercial fishery value per 1 m2. This approach provides a snapshot of the use of these reefs by commercial fish species; other species such as prey species likely contribute to the commercial fishery value of the reefs, but we assume that this value is integrated through this snapshot approach.

We were unable to recover all tray samples due to logistical difficulties and therefore data only exist for one year, or 4 seasons, for resident fauna; a total of 99 substrate trays out of the initial 180 were recovered and sampled (Summer 2009 = 23; Fall 2009 = 31; Winter 2009 = 25; Spring 2010 = 13; Summer 2010 = 7). This allowed us to calculate resident nekton biomass and commercial fishery value with one year of data; however, to calculate values for the duration of our study, we used these same data for subsequent seasons under the assumption that the reef and mud-bottom treatments would produce at least as much as during the first year of sampling. While this assumption may lead to slightly conservative estimates of nekton biomass, it has been shown in this particular estuary that resident fish and invertebrate biomass at oyster reefs does not increase significantly from one year to the next (Humphries et al., 2011).

To calculate the cost of constructing the oyster reefs, we used the costs from labor and materials and divided it by the total area of reef created to get a cost per 1 m2. We assumed no costs to maintain reefs.

Statistical analyses

All environmental parameters and nekton data were tested with three-way analysis of variance (ANOVA) where date, region, and treatment were treated as fixed effects. Nekton biomass (kg m−2) was modeled separately for each gear type. Where there were interactions in ANOVAs, we used main effects models with linear contrasts to determine formal relationships. Data were tested for normality using Shapiro–Wilks test and quantile–quantile plots, and no transformations were necessary.

We analyzed nekton taxa individually to evaluate treatment effects on the most common species, i.e., species that were more than 3% of the total catch. We used individual pairwise t-tests to compare each species’ mean overall biomass between the paired oyster reef and mud-bottom treatments.

We modeled nekton biomass and commercial fishery values separately with one-way ANOVAs and linear regression, using date as the fixed effect. Normality of data was assessed using quantile–quantile plots. All statistical analyses were done in R, version 3.1.3 (R Core Development Team, 2015).

Results

Environmental variables

Temperature and dissolved oxygen mean and variation showed typical seasonal patterns and were not significantly different among regions or treatment at any date (Table 1). Mean daily water temperature was 23.8 °C (±0.4) and ranged between 5 and 35 °C, and mean dissolved oxygen was 7.6 mg L−1 (±0.1) and ranged between 2 and 16 mg L−1 throughout the experiment. Salinity was greater at the South region during all years (p < 0.5) with a mean of 13.1 (±0.6) as compared to means of 9.5 (±0.5) (West) and 10.2 (±0.6) (North), with an overall range of 0–23. Similarly, Secchi depth was greater at South region during all years (p < 0.05) with a mean of 47.6 cm (±1.9) as compared to means of 38.7 cm (1.6) (West) and 41.3 cm (2.1) (North) and overall ranged between 10 and 70 cm.

Table 1 Environmental parameters.

Mean (±standard error) of discrete water quality samples collected at each site during each sample event (n = 627). Letters after means indicate significant differences between regions within years (p < 0.05).

	Year 1	Year 2	Year 3	
	North	South	West	North	South	West	North	South	West	
Temperature (°C)	26.4 (0.7)	24.8 (0.8)	25.1 (0.7)	24.1 (1.3)	23.2 (1.3)	23.3 (1.4)	22.9 (1.1)	22.5 (1.3)	22.3 (1.4)	
Dissolved oxygen (mg L−1)	7.5 (0.2)	7.4 (0.3)	7.4 (0.2)	6.7 (0.4)	6.4 (0.4)	7.6 (0.6)	8.2 (0.4)	8.5 (0.6)	8.9 (0.5)	
Salinity	9.3 (0.6)a	13.3 (0.7)b	8.9 (0.7)a	9.5 (0.7)a	12.1 (0.6)b	9 (0.7)a	11.7 (0.9)	13.9 (1.1)	11.1 (0.9)	
Secchi depth (cm)	35 (2.1)a	42.4 (2.2)b	37.1 (1.3)a	47.9 (2.6)a	49 (1.6)a	36.1 (2.3)b	41.8 (2.4)a	55.1 (3.6)b	47 (2.3)a	

Nekton biomass

In 120 gill net samples a total of 1,805 individuals from 32 species, resulting in 749.28 kg were collected . Oyster reefs had a mean biomass of 0.155 (±0.019) kg m−2, and mud-bottom was 0.09 (±0.018) kg m−2. In 240 seine samples, a total of 17,968 individuals were collected that represented 54 species and 14.86 kg total, with a mean biomass of 0.001 (±0.001) kg m−2 at both oyster reefs and mud-bottom. The 99 tray samples resulted in 1.33 kg of fish and invertebrate biomass from 1,592 individuals and 21 total species. Oyster reefs had a mean biomass of 0.05 (±0.08) kg m−2, whereas mud-bottom was 0.008 (±0.002) kg m−2.

ANOVA results from the gill net (transients) and tray data (residents) showed a significant treatment and date effect (Table 2 and Fig. 2). Individual contrasts indicated that gill nets on oyster reefs in Fall 2010 and Spring 2011 had significantly higher biomass values than mud-bottom reference sites. All comparisons for tray data between oyster reefs and mud-bottom sites were statistically significant. ANOVA results for the seine data were all non-significant except for date.

Figure 2 Total nekton biomass.

Box and whisker plots of nekton biomass (kg m−2) of species caught in (A) gill net, (B) seine, and (C) tray gears at experimental oyster reef and paired reference mud-bottom sites. Significant differences between reef and mud-bottom reference values are indicated by an asterisk (p < 0.05).

Table 2 Statistical results.

Results from three-way analysis of variance (ANOVA) on nekton biomass data. Treatment refers to reef and mud-bottom habitat. Bold values indicate statistical significance at the p < 0.05 level.

	Df	F	p-value	
Gill net	
Date	9	2.155	0.039	
Region	2	2.068	0.089	
Treatment	1	8.002	0.006	
Date × Region	18	1.566	0.100	
Date × Treatment	9	1.134	0.354	
Region × Treatment	2	2.329	0.091	
Date × Region × treatment	18	1.167	0.317	
Seine	
Date	9	4.349	<0.001	
Region	2	0.435	0.650	
Treatment	1	0.004	0.949	
Date × Region	18	0.802	0.690	
Date × Treatment	9	0.725	0.684	
Region × Treatment	2	0.495	0.612	
Date × Region × Treatment	18	0.579	0.901	
Tray	
Date	3	4.221	0.021	
Region	2	1.936	0.175	
Treatment	1	20.243	<0.001	
Date × Region	6	0.455	0.832	
Date × Treatment	3	1.725	0.200	
Region × Treatment	2	1.794	0.196	
Date × Region × Treatment	3	0.778	0.523	

All species except Northern puffer (Sphoeroides maculatus) collected in gill nets had higher biomass at oyster reefs than mud-bottom (Table 3 and Fig. 3). However, only sheepshead (Archosargus probatocephalus) was statistically significant (p < 0.01). In seines, no species showed significant differences in pairwise t-tests, although Gulf menhaden (Brevoortia patronus), brown shrimp (Penaeus aztecus), blue crab (Callinectes sapidus), and bay anchovy (Anchoa mitchilli) all had higher biomass values at oyster reefs. All species except blue crab had higher biomass at oyster reefs in tray samples, and four species were statistically significant: speckled eel (Myrophis punctatus), mud crab (Panopeus herbstii), naked goby (Gobiosoma bosc), and freckled blenny (Hypsoblennius ionthas) (p < 0.05). One exception to this trend was the blue crab, which had greater biomass at mud-bottom (p < 0.01).

Figure 3 Nekton species biomass.

Mean (±one standard error) nekton biomass (kg m−2) of dominant (i.e., >3% of total biomass) species captured with (A) gill net, (B) seine, and (C) trays at at all experimental oyster reef and paired reference mud-bottom sites. Significant differences between reef and mud-bottom reference values are indicated by an asterisk (p < 0.05).

Table 3 Nekton species captured in study.

Mean (±one standard error) increased nekton biomass ($ kg m−2), commercial fish price ($ kg−1), and increased value of commercial fishery (m−2) by species and gear type from experimental oyster reef versus paired reference mud-bottom sites. Only dominant species (i.e., >3% of total biomass) are presented. Negative values indicate mud-bottom references sites had a greater value for that particular year. All currency is in 2011 US dollars.

Gear	Species	Common name	Increased biomass (kg m−2)	Commercial fish price ($ kg−1)	Increased fishery value ($ m−2)	
			Year 1	Year 2	Year 3		Year 1	Year 2	Year 3	
	Archosargus probatocephalus	Sheepshead	0.0216 (0.0144)	0.0106 (0.0096)	0.0334 (0.0117)	1.26	0.027 (0.018)	0.013 (0.012)	0.042 (0.015)	
	Arius felis	Hardhead catfish	0.0006 (0.0051)	0.0108 (0.0068)	0.0051 (0.0064)	1.05	0.001 (0.005)	0.011 (0.007)	0.005 (0.007)	
	Bagre marinus	Gafftopsail catfish	−0.0033 (0.0026)	0.0023 (0.0053)	0.0001 (0.0003)	1.05	−0.003 (0.003)	0.002 (0.006)	0 (0)	
	Brevoortia patronus	Gulf menhaden	0.0072 (0.0037)	0.0003 (0.0005)	0.0005 (0.0003)	0.17	0.001 (0.001)	0 (0)	0 (0)	
	Callinectes sapidus	Blue crab	−0.0003 (0.0009)	0.0007 (0.0006)	0 (0)	2.01	−0.001 (0.002)	0.001 (0.001)	0 (0)	
	Caranx hippos	Crevalle jack	0 (0)	0 (0)	−0.0008 (0.0005)	2.02	0 (0)	0 (0)	−0.002 (0.001)	
	Carcharhinus leucas	Bull shark	0.0223 (0.0139)	0.0151 (0.0089)	−0.0148 (0.0149)	1.80	0.04 (0.025)	0.027 (0.016)	−0.027 (0.027)	
	Chaetodipterus faber	Atlantic spadefish	0.0002 (0.0002)	0.0008 (0.0006)	0.0004 (0.0003)	0.16	0 (0)	0 (0)	0 (0)	
	Cynoscion nebulosus	Spotted seatrout	0.0032 (0.0024)	−0.0026 (0.0034)	0.0165 (0.0075)	4.53	0.014 (0.011)	−0.012 (0.016)	0.075 (0.034)	
	Dasyatis sabina	Atlantic stingray	0 (0)	0.0012 (0.0012)	0 (0)	0.45	0 (0)	0.001 (0.001)	0 (0)	
Gill net	Dorosoma cepedianum	Gizzard shad	0 (0)	−0.0001 (0.0003)	−0.0003 (0.0008)	1.74	0 (0)	0 (0.001)	0 (0.001)	
	Elops saurus	Ladyfish	0.0007 (0.0007)	0.0001 (0.0011)	0.0006 (0.001)	0.00	0 (0)	0 (0)	0 (0)	
	Lagodon rhomboides	Pinfish	0 (0)	0 (0)	−0.0001 (0.0001)	1.29	0 (0)	0 (0)	0 (0)	
	Leiostomus xanthurus	Spot	0.0001 (0.0001)	0 (0.0001)	0 (0)	1.84	0 (0)	0 (0)	0 (0)	
	Micropogonias undulatus	Atlantic croaker	0.0022 (0.0015)	0.0002 (0.0004)	−0.0004 (0.0033)	1.67	0.004 (0.003)	0 (0.001)	−0.001 (0.005)	
	Mugil cephalus	Striped mullet	0.0011 (0.0021)	0.0052 (0.0021)	0.0015 (0.002)	1.52	0.002 (0.003)	0.008 (0.003)	0.002 (0.003)	
	Oligoplites saurus	Skipjack	0 (0)	−0.0009 (0.001)	0.0006 (0.0006)	2.02	0 (0)	−0.002 (0.002)	0.001 (0.001)	
	Paralichthys lethostigma	Southern flounder	0.0022 (0.0033)	0.003 (0.003)	0.0075 (0.0032)	4.45	0.01 (0.015)	0.013 (0.013)	0.033 (0.014)	
	Pogonias cromis	Black drum	0.023 (0.038)	0.0439 (0.0235)	0.0297 (0.0304)	0.00	0 (0)	0 (0)	0 (0)	
	Sciaenops ocellatus	Red drum	0.0037 (0.0158)	0.0118 (0.0063)	−0.0006 (0.0045)	0.00	0 (0)	0 (0)	0 (0)	
	Sphoeroides maculatus	Northern puffer	0 (0)	−0.0237 (0.0273)	0 (0.0007)	0.00	0 (0)	0 (0)	0 (0)	
	Anchoa mitchilli	Bay anchovy	0.0001 (0.0001)	0.0001 (0.0002)	−0.0001 (0.0001)	0.25	0 (0)	0 (0)	0 (0)	
	Bairdiella chrysoura	Silver perch	0 (0)	−0.0001 (0.0001)	0.0001 (0.0001)	1.84	0 (0)	0 (0)	0 (0)	
	Brevoortia patronus	Gulf menhaden	0.0007 (0.0007)	0.0002 (0.0005)	0 (0)	0.17	0 (0)	0 (0)	0 (0)	
	Callinectes sapidus	Blue crab	0.0002 (0.0002)	0 (0)	0 (0)	2.01	0 (0)	0 (0)	0 (0)	
	Dasyatis sabina	Atlantic stingray	−0.0001 (0.0001)	0 (0)	0 (0)	0.45	0 (0)	0 (0)	0 (0)	
	Lagodon rhomboides	Pinfish	0 (0)	0 (0)	−0.0001 (0.0001)	1.29	0 (0)	0 (0)	0 (0)	
Seine	Menidia beryllina	Inland silverside	0 (0)	0 (0)	0.0001 (0.0001)	0.00	0 (0)	0 (0)	0 (0)	
	Micropogonias undulatus	Atlantic croaker	−0.0001 (0.0001)	0 (0)	0 (0)	1.67	0 (0)	0 (0)	0 (0)	
	Mugil cephalus	Striped mullet	0.0001 (0.0001)	−0.0001 (0.0001)	0 (0)	1.52	0 (0)	0 (0)	0 (0)	
	Palaemonetes pugio	Grass shrimp	−0.0001 (0.0001)	0.0001 (0)	0 (0)	0.00	0 (0)	0 (0)	0 (0)	
	Penaeus aztecus	Brown shrimp	0 (0.0002)	0.0001 (0.0001)	0.0002 (0.0002)	4.34	0 (0.001)	0.001 (0)	0.001 (0.001)	
	Penaeus setiferus	White shrimp	0 (0)	0 (0)	−0.0001 (0.0001)	4.34	0 (0)	0 (0)	0 (0)	
	Sciaenops ocellatus	Red drum	−0.0002 (0.0002)	0 (0)	0 (0)	0.00	0 (0)	0 (0)	0 (0)	
Tray	Alpheus heterochaelis	Bigclaw snapping shrimp	0 (0)	0.0001 (0.0001)	NA	0.00	0 (0)	0 (0)	NA	
Anchoa mitchilli	Bay anchovy	−0.0004 (0.0004)	0 (0)	NA	0.25	0 (0)	0 (0)	NA	
Callinectes sapidus	Blue crab	−0.0017 (0.0011)	−0.0009 (0.0009)	NA	2.01	−0.003 (0.002)	−0.002 (0.002)	NA	
Chasmodes bosquianus	Stripped blenny	0.0002 (0.0002)	0 (0)	NA	0.00	0 (0)	0 (0)	NA	
Cynoscion nebulosus	Spotted seatrout	−0.0004 (0.0004)	0 (0)	NA	4.53	−0.002 (0.002)	0 (0)	NA	
Gobiesox strumosus	Skilletfish	0.0031 (0.0014)	0 (0)	NA	0.00	0 (0)	0 (0)	NA	
Gobiosoma bosc	Naked goby	0.0046 (0.0015)	0.0011 (0.0011)	NA	0.00	0 (0)	0 (0)	NA	
Hypsoblennius ionthas	Freckled blenny	0.0053 (0.0027)	0.0003 (0.0003)	NA	0.00	0 (0)	0 (0)	NA	
Mugil cephalus	Striped mullet	−0.0001 (0.0001)	0 (0)	NA	1.52	0 (0)	0 (0)	NA	
Myrophis punctatus	Speckled worm eel	0.0023 (0.001)	0.0004 (0.0004)	NA	0.00	0 (0)	0 (0)	NA	
Opsanus beta	Gulf toadfish	0.0055 (0.0045)	0 (0)	NA	0.00	0 (0)	0 (0)	NA	
Palaemonetes pugio	Grass shrimp	−0.0014 (0.0007)	0.0005 (0.0004)	NA	0.00	0 (0)	0 (0)	NA	
Panopeus herbstii	Atlantic mud crab	0 (0)	0.0012 (0.0012)	NA	0.00	0 (0)	0 (0)	NA	
Penaeus aztecus	Brown shrimp	0.0001 (0.0001)	0 (0)	NA	4.34	0.001 (0)	0 (0)	NA	
Penaeus setiferus	White shrimp	−0.0001 (0.0001)	0 (0)	NA	4.34	0 (0)	0 (0)	NA	
Rhithropanopeus harrisii	Harris mud crab	0.0277 (0.0057)	0.023 (0.0077)	NA	0.00	0 (0)	0 (0)	NA	

Nekton biomass at oyster reefs was 212% greater than at mud-bottom, or 0.122 (±0.039) kg m−2 (Table 4). The potential for oyster reefs to increase nekton biomass was positive in all sampling periods except one (Fig. 4A). The linear model, however, was not statistically significant (F = 1.133, p-value = 0.358) and had an r-squared value of 0.169, with an intercept of 0.11 and slope of −0.003.

Figure 4 Increased nekton biomass and value.

Mean (±one standard error) of increased (A) nekton biomass ($ kg m−2) and (B) commercial fishery value ($ m−2; 2011 US dollar) of total nekton catch at all oyster reef sites as compared to reference, mud-bottom sites. Linear model results are displayed in the top corner.

Table 4 Increased nekton biomass and value.

Mean (±one standard error) nekton biomass (kg m−2) and value of commercial fishery ($ m−2) from experimental oyster reef versus paired reference mud-bottom sites by sampling year. Overall mean (±one standard error) values are presented in the last row for the entire study period. Reef cost is a one time, initial cost that requires no maintenance through time (La Peyre et al., 2014).

Year	Oyster reef nekton biomass (kg m−2)	Mud-bottom nekton biomass (kg m−2)	Oyster reef nekton value ($ m−2)	Mud-bottom nekton value ($ m−2)	Oyster reef cost ($ m−2)	
1	0.2641 (0.0323)	0.1219 (0.0309)	0.15 (0.04)	0.08 (0.01)	105.03	
2	0.1783 (0.0439)	0.0715 (0.0453)	0.18 (0.07)	0.05 (0.03)	0	
3	0.2132 (0.0183)	0.0955 (0.0051)	0.17 (0.03)	0.12 (0.02)	0	
Mean	0.2185 (0.0315)	0.0963 (0.0271)	0.17 (0.04)	0.08 (0.02)	35.01	

Commercial fishery value

Commercial fish prices of species captured ranged from $0 to $4.53 kg−1 (2011 dollars; Table 3). Species that contributed most to augmenting the potential commercial fishery value surrounding restored oyster reefs were sheepshead, spotted seatrout (Cynoscion nubulosus), Southern flounder (Paralichthys lethostigma), and bull shark (Carcharhinus leucas), all of which were caught with gill nets (Table 2).

Potential commercial fishery value at oyster reefs was more than double than at mud-bottom, or $0.09 (±$0.06) m−2 (Table 4). The potential for oyster reefs to increase the local commercial fishery value was positive in all sampling periods except two (Fig. 4B). The linear model was statistically significant (F = 2.208, p-value = 0.037) and had an r-squared value of 0.284, with an intercept of −0.003 and slope of 0.02.

The cost to build all the reefs in 2009 was $23,545.23 (or $24,686.84 in 2011 dollars). With a total reef area of 235 m2, this equates to approximately $105.03 to build 1 m2 of reef with unaggregated oyster shell (Table 4). This is a one-time, initial cost, and we assume no maintenance is necessary for these reefs to persist and be self-sustaining because of high recruitment and low oyster mortality rates on these reefs (La Peyre et al., 2014).

Discussion

The restored oyster reefs provided increased biomass of nekton and supported augmented commercial fisheries value. In both cases, the reefs more than doubled the provision of the service as compared to reference sites. In real numbers, the actual augmentation values were relatively low (0.12 kg m−2, $0.09 m−2). This likely reflects reef location within a complex of extensive oyster beds and productive coastal marsh (Grabowski et al., 2005), highlighting the need to understand the local context of any restoration project. Further analyses to quantify recreational fishery value and other provided ecosystem services would contribute to more complete analyses of local reef restoration benefits.

The hypothesis that restoration of estuarine habitats may increase nekton biomass and commercial fishery value relies on the assumption that either quality or quantity of habitat is limiting (Turner et al., 2000). For oysters, restoration studies often assume reef or structured habitat is limiting because of global declines in functional oyster reef habitat (Peterson, Grabowski & Powers, 2003; Beck et al., 2011; Grabowski et al., 2012). However, due to a lack of historical and current data, it is unclear if Louisiana suffers significant declines of oyster habitat. Furthermore, our study area has been a public oyster seed reservation since 1940, and almost 30% of the bottom in the area is classified as oyster beds (Louisiana Department of Wildlife and Fisheries, 2011). Despite the extensive oyster reef habitat in the area, and the productive salt marshes surrounding our created reefs, our data still capture an overall increase in nekton biomass. This finding suggests that either fish and invertebrate biomass still remains limited by habitat availability despite the extensive reefs already present, or that nekton may be limited by the quality or characteristics of the habitat available.

Although vegetated habitats and oyster reefs are not functionally equivalent, they may provide similar services (Geraldi et al., 2009). For instance, we found that the restored oyster reefs did not affect facultative resident species, or individuals that are potentially using the reef during an ontogenetic shift. It is possible that while the reefs provided added structural habitat, the marginal effect was relatively low to the adjacent marsh edges. In fact, our restored reefs and reference sites were all adjacent to marsh edge habitat, which has been shown to provide valuable habitat for numerous similar nekton species (Stunz, Minello & Rozas, 2010). Within the marsh and shallow water estuarine complexes of coastal Louisiana, determining which habitats may limit or support organisms can be difficult (Baltz, Rakocinski & Fleeger, 1993; Chesney, Baltz & Thomas, 2000).

Habitat quality affects species use. For oyster reefs, quality may be considered as reef size, height, and material base, all of which have been shown to influence ecosystem services (Lenihan, 1999; Lenihan et al., 2001). These structural characteristics may explain the increase in resident fish and invertebrate species biomass we observed in this study. For example, the existing reef system in our study area consists predominantly of flat, harvested subtidal structures, while our restored reefs were three-dimensional intertidal habitats. The difference in habitat morphology is potentially driving species use (Humphries et al., 2011), and our results further corroborate this hypothesis. A caveat to this, however, is that our restored reefs provided immediate support of nekton biomass, which did not significantly increase over time. This is despite significant increases in structural complexity via recruitment and growth of oysters (Casas, La Peyre & La Peyre, 2015). This finding suggests that simply the existence of three-dimensional structure, and the lack of hard bottom rather than any reef characteristics may be a limiting factor.

The immediate support of higher nekton biomass on the reefs as compared to mud-bottom habitat suggests that the higher reef biomass initially resulted from a shift in the local population from surrounding habitats. In restoring habitats for nekton, the question of whether the new habitat is simply attracting individuals from the local populations, or actually enhancing production remains key, and difficult in valuing the restored habitat (e.g., Grossman, Jones & Seaman, 1997; NOAA, 1997). Generally, evidence for enhancement of production involves determining whether the restored habitat provides habitat that limits species recruitment, and secondly, whether new reefs provide spatial refuge from predation and increased prey resources i.e., Powers et al., 2003; Peterson, Grabowski & Powers, 2003. Previous work has assumed that species with enhanced recruits on reefs relative to reference sediments are limited by current reef area (Powers et al., 2003; Peterson, Grabowski & Powers, 2003). While our sampling documented a range of resident and transient fish sizes, few recruit-sized individuals were captured in either the reef or the mud-bottom. However, if the reef provides enhanced reef-associated prey resources, as indicated by our data with the increased resident species biomass, it would support the contention that the reef is enhancing fish production, not through the addition of fish to the local population, but by increasing survival and enhancing growth of individuals already in the regional population. Enhanced densities of fish (transient, resident of all size classes) thus reflect positive impacts of the reef on survival (Powers et al., 2003). For resident species specifically, which generally have small home ranges (Teo & Able, 2003; Potthoff & Allen, 2003), the assumption would be that this new habitat offered better refuge from predation (Hixon, 1998; Humphries, La Peyre & Decossas, 2011), resulting in enhanced survival, and thus production.

Our restored oyster reefs were estimated to provide the local commercial fishery with $0.17 m−2. This value failed to cover the costs of construction within the three years of monitoring. Other studies have reported higher values (e.g., $0.41 m−2 by Grabowski et al., 2012) however, the differences in numbers likely reflect location- or region-specific effects of reefs, as it is difficult to transfer values from ecosystem services out of their original context or region (Turner et al., 2003). Additionally, black (Pogonias cromis) and red drum (Sciaenops ocellatus) contributed 0.03 and 0.01 kg m−2, respectively, to the overall increased nekton biomass value of 0.12 kg m−2. These two drum have no commercial value in Louisiana, but high recreational value which is significant in the area. For instance, in 2006, the recreational fishery value in coastal Louisiana was estimated to augment local economies by over $750 million and support nearly 8,000 jobs (Coastal Restoration and Protection Authority of Louisiana, 2012). Overall, however, this means that only 34% of the nekton biomass was factored into the calculation of commercial fishery value. Given the high value of recreational fishing in many coastal areas, developing methods to include their value would contribute to more accurate valuation of reef services.

Alone, the potential increased commercial fishery value of the restored oyster reefs fail to cover the costs of construction within a three-year time frame. This snapshot approach using biomass fails to capture actual production support, which would include accounting for recruitment and future reef productivity (Peterson, Grabowski & Powers, 2003). As a result, these results likely underestimate the full value of the reef. Additionally, full valuation of restoration activities could include other services such as shoreline protection, recreational fisheries enhancement and water quality improvements, which have been estimated to be as much as $17,836 per hectare (Grabowski et al., 2012). The valuation of services and relative value of different functions likely varies greatly between locales and regions. Given the land loss and hypoxia issues coastal Louisiana faces, other services such as shoreline protection and water quality improvement may provide a higher valuation estimate in comparison to other regions.

Supplemental Information

Supplemental Information 1 Dataset for mansucript

Click here for additional data file.

We thank P Yakupzack, C Duplechain, G Decossas, S Miller, A Catalenello, S Beck, T Mace, L Broussard, S Hein, P Banks, H Finley, S Casas-Liste, J Gordon, J La Peyre, L Schwarting, A Honig, and B Eberline for providing logistical, field, and laboratory assistance. Thanks to M Voisin and Motivatit Oysters for providing oyster shell. We also thank J Grabowski, M Griffin, and 2 anonymous reviewers for making comments that greatly improved earlier versions of the manuscript. Any use of trade, product, or firm names is for descriptive purposes only and does not imply endorsement by the US Government.

Additional Information and Declarations

Competing Interests

Author Contributions

Animal Ethics

The authors declare there are no competing interests. Any use of trade, product, or firm names is for descriptive purposes only and does not imply endorsement by the US Government.

Austin T. Humphries and Megan K. La Peyre conceived and designed the experiments, performed the experiments, analyzed the data, contributed reagents/materials/analysis tools, wrote the paper, prepared figures and/or tables, reviewed drafts of the paper.

The following information was supplied relating to ethical approvals (i.e., approving body and any reference numbers):

We obtained all necessary permits for the described study and operated under a scientific collection permit from the Louisiana Department of Wildlife and Fisheries to Dr. Megan La Peyre (S-03-2009, S-105-OYS-2010). No endangered or protected species were collected during this project. Furthermore, fish and invertebrates were collected under the Institutional Animal Care and Use Committee permit 08-005 to Dr. Megan La Peyre through the Louisiana State University Institutional Animal Care and Use Committee.

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
