# Peer review of "Oyster reef restoration supports increased nekton biomass and potential commercial fishery value"

_PeerJ, doi:10.7717/peerj.1111_

## Round 0.1 · original submission · Major Revisions

Please, amended the manuscript following the reviewer suggestions.

Reviewer 1 ·

Basic reporting

The paper is generally well written and provides good justification for the study.

Experimental design

This is the main problem I see with the manuscript. The study should have been run using a 3-way ANOVA (site, date, trt) or a blocked ANOVA using site as a blocking factor. Running a bunch of pairwise t-tests is a classic mistake of not controlling experimentwise error rate. They would need to apply a Bonferroni adjustment (dividing the alpha level (0.05) by the number of comparisons) to get the correct rejection rate. You aren’t allowed to “ignore the interactive effects” as stated in line 212. Instead, the 3-way or blocked ANOVA should be run first, and if there are interactions that limit interpretation, then a simple main effects model would need to be run and linear contrasts could be used. The statistics should be re-run using these models.

Validity of the findings

The data will stay the same, but interpretation of statistical significance may change after the statistical analyses are corrected.

Additional comments

The paper is generally well written and provides good justification for the study. However, some of the statistical analyses were not done correctly and should be re-run in order to obtain correct rejection rates (details below under “statistical analysis”).

Other specific comments:
Line 107-114: I suggest the text be modified to clarify the locations of the study site on the map. The text describes 3 study areas (maybe call these regions?) on the map – North, West, and South. Those are clear. The text then describes “Within each sample, two 2-25 m shoreline segments were identified…one was designated as the reference mud bottom site and the other as the reef site” – are these the #1 and #2 points on the map? Or are there 2-25 m shoreline segments (one mud bottom and one reef) at each #1 and #2 point on the map?
Line 108: Delete “but” before 200 m
Line 133: Were the gill nets set at a certain time / tide?
Line 142: Why were the mud bottom trays sampled with replacement, but the reef trays sampled without replacement? I would be concerned there is a different disturbance regime being encountered by each community, which may affect successional patterns and results.
Line 150: I am concerned that the sampling activity (seines and tray removal) may have affected the gill net results by scaring away transient species, for example. Was there any test of this?
Line 159-160: Before the biomass data were divided by a particular reef’s area, were they scaled to the area sampled by the individual gear type?
Line 165: Add the phrase “across gear types”
Line 173-175: Were any species collected in multiple gear types? Was the calculation adjusted so as not to double count or double value these species?
Line 175: Add the phrase “across gear types”
Line 183: delete “s” from “exists”
Statistical analysis: This is the main problem I see with the manuscript. The study should have been run using a 3-way ANOVA (site, date, trt) or a blocked ANOVA using site as a blocking factor. Running a bunch of pairwise t-tests is a classic mistake of not controlling experimentwise error rate. Essentially, they would need to apply a Bonferroni adjustment by dividing the alpha level (0.05) by the number of comparisons to get the correct rejection rate. You aren't allowed to “ignore the interactive effects” as stated in line 212. Instead, the 3-way or block ANOVA should be run first, and if there are interactions that limit interpretation, then a simple main effects model would need to be run and linear contrasts could be used. The statistics should be re-run using these models.
Line 235: Begin this sentence with “In 120 gill net samples”
Line 240-241: Table 1 is referenced in the statement that oyster reefs had a mean biomass of 0.05 kg m-2 compared to 0.008 kg m-2 on mud bottom, but these data do not appear in this table that I can see.
Line 243: Add “transients on” oyster reefs
Line 259: What re the equation and parameter values of the linear model? (same question for line 269-270)
Line 281: The statement that this observation “likely reflects the reefs location…” needs a reference.
Line 294: Is it possible this increase in nekton biomass is because organisms are moving from one habitat area to the restored area (i.e. spreading out rather than increasing production?). Do you have any data from these surrounding reef and marsh habitats in terms of abundance/biomass of nekton?
Line 322-324: I think the authors are on to something here – is there a way to also calculate recreational fishing value, which may be significant for certain species? This is a great idea and could be expanded, perhaps even with a back of the envelope calculation in the discussion.
Figure 1: The figure caption denotes the location of 6 experimental oyster reefs and 6 mud bottom sites, but there are only 6 points on the map. This should be clarified.
Figure 2: For the boxplots, the identity of the dots (outliers?) should be explained. Without paying close attention, they look like small asterisks. I suggest labeling these 2 figures as (A), (B), (C) for easier referencing in the text. The y-axis title for the “residents” graph is jumbled together. In the text, these graphs are sometimes referred to by their gear type (gill net, bag seine, tray) rather than their nekton identity – I suggest the gear type be incorporated within each figure title or explained in the figure caption.
Figure 3: Same partial comment as above: I suggest labeling these 2 figures as (A), (B), (C) for easier referencing in the text. Also, in the text, these graphs are sometimes referred to by their gear type (gill net, bag seine, tray) rather than their nekton identity – I suggest the gear type be incorporated within each figure title or explained in the figure caption.
Figure 4: I suggest labeling these 2 figures as (A) & (B) for easier referencing in the text. The y-axis titles for both figures are jumbled together. Suggest including the equation for the linear model on the figures.
Table 1: First sentence of the caption – is this calculated as (reef-mud)? If so, add a phrase that explains that here. What happens when biomass or fishery value are negative? Do the negative numbers get carried forward when summing across reef types, or is the negative number replaced with a zero?
Table 2: “Oyster” is misspelled in the last column (Oyster reef cost).

Reviewer 2 ·

Basic reporting

The structure of this manuscript (ms) is consistent with accepted standards, except that some of the labels of Figures 2 and 4 are jumbled.

Experimental design

I have integrated my comments for "Experimental Design" and "Validity of the Findings" into the paragraphs below.

The questions were justified and posed clearly. However, the experimental design was not appropriate for the main question, which was to provide “quantitative estimates of fisheries enhancement by oyster reef restoration.” This question has been posed previously and numerous times in the context of artificial reefs, as whether artificial reefs enhance secondary production or merely concentrate abundance of reef-associated species. To test these alternative hypotheses one has to demonstrate that biomass or abundance is enhanced at the level of the population, not just at a local level. The present study has demonstrated enhanced biomass and abundance of some species locally, which could be due to a simple shift across habitats, not population enhancement. The data presented in the ms actually suggests that the local enhancement was merely a shift in habitat use, because the biomass and abundance increases happened very quickly, which could only be due to immigration as opposed to long-term recruitment. Consequently, any statements that these reefs enhanced fishery biomass and value must be deleted. The data only suggest that there may be enhancement of production and value. The authors note the problem with enhancement vs. concentration (lines 285-296 and 313-314), but then ignore it when drawing the main conclusion.

Another issue with the experimental design was with the use of trays in mud bottom. Were these truly representative of the benthic community in mud? I suspect not because the trays only went 11 cm into the sediment, and many long-lived high-biomass species dwell deeper than 11 cm, such that the faunal estimates for mud bottom are likely to be biased low. Had the authors used deep suction/core samples to validate the tray samples, I would be much more confident in the results. At this time, without suction/core samples, the bias of the trays in mud bottom is unknown, which renders the comparisons with oyster reef trays unreliable.

Regarding local biomass enhancement, most of the species-specific comparisons between mud bottom and oyster reef were non-significant, and should be removed from the estimates of total biomass.

Overall, this study shows evidence for habitat use of oyster reefs and mud bottom, definitely not for enhancement of secondary production and fishery value by oyster reefs relative to mud bottom.

Validity of the findings

See paragraphs in the "Experimental Design" section above.

Additional comments

I strongly urge the authors to revise the ms to deal with habitat use, not population enhancement, of oyster reefs. Better to revise it now than to have it criticized publicly in the literature.

·

Basic reporting

no comments

Experimental design

no comments

Validity of the findings

no comments

Additional comments

The paper was laid out well and provides a good snapshot of value of oyster restoration within a local context. I have a few minor comments - please see attached document.

---

## Round 0.2 · accepted · Accept

The authors have improved the paper according to the suggestions made by reviewers.

Reviewer 1 ·

Basic reporting

Sufficient improvements have been made to the basic reporting, in particular as it relates to enhanced fish production.

Experimental design

Statistics have been revised appropriately.

Validity of the findings

I am satisfied that findings are valid.

Reviewer 2 ·

Basic reporting

See my earlier review.

Experimental design

See my earlier review.

Validity of the findings

See my earlier review.

Additional comments

The revised manuscript has addressed my previous criticisms.